# Extraction and Analysis of Finer Impervious Surface Classes in Urban Area

**Wenyue Liao** [1,2,†], **Yingbin Deng** [2,3,†], **Miao Li** [1,*] 🄳, **Meiwei Sun** [1,2], **Ji Yang** [2,3] and **Jianhui Xu** [2,3]

1 College of Geographical Science, Harbin Normal University, Harbin 150025, China; wy_liao@hrbnu.edu.cn (W.L.); mwsun@hrbnu.edu.cn (M.S.)
2 Guangdong Open Laboratory of Geospatial Information Technology and Application, Laboratory of Guangdong for Utilization of Remote Sensing and Geographical Information System, Guangzhou Institute of Geography, Guangdong Academy of Sciences, Guangzhou 510070, China; yingbin@gdas.ac.cn (Y.D.); yangji@gdas.ac.cn (J.Y.); xujianhui306@gdas.ac.cn (J.X.)
3 Southern Marine Science and Engineering Guangdong Laboratory (Guangzhou), Guangzhou 511485, China
* Correspondence: mli@hrbnu.edu.cn
† Both authors contributed equally.

**Abstract:** Impervious surfaces (IS), the most common land cover in urban areas, not only provide convenience to the city, but also exert significant negative environmental impacts, thereby affecting the ecological environment carrying capacity of urban agglomerations. Most of the current research considers IS as a single land-cover type, yet this does not fully reflect the complex physical characteristics of various IS types. Therefore, limited information for urban micro-ecology and urban fine management can be provided through one IS land-cover type. This study proposed a finer IS classification scheme and mapped the detailed IS fraction in Guangzhou City, China using Landsat imagery. The IS type was divided into seven finer classes, including blue steel, cement, asphalt, other impervious surface, and other metal, brick, and plastic. Classification results demonstrate that finer IS can be well extracted from the Landsat imagery as all root mean square errors (RMSE) are less than 15%. Specially, the accuracies of asphalt, plastic, and cement are better than other finer IS types with the RMSEs of 7.99%, 8.48%, and 9.92%, respectively. Quantitative analyses illustrate that asphalt, other impervious surface, and brick are the dominant IS types in the study area with the percentages of 9.68%, 6.27%, and 4.45%, respectively, and they are mainly located in Yuexiu, Liwan, Haizhu, and Panyu districts. These results are valuable for research into urban fine management and can support the detailed analysis of urban micro-ecology.

**Keywords:** subpixel classification; impervious surface; urban environment; finer IS category

## 1. Introduction

Large numbers of natural ground objects such as vegetation and soil located in cities are being replaced by buildings made of asphalt, colored steel, cement, and other materials. This has a negative impact on urban water resources, local climate, living environment, etc. [1]. Such man-made substances are denoted as impervious surfaces (IS) as they cannot be penetrated by water, and related land-cover variations are crucial in research on urban expansion and urban land use changes [2]. The detailed classification of impervious surfaces not only reflects the subtle changes in the development of urban built-up areas at the micro level, but also provides key information for city governments in order to make effective management and planning decisions.

In the sub-pixel scale land-use classification [3–5], the IS class is often represented by high and low reflectivity objects [6–8], and bright and dark objects [9]. Phinn et al. [4] used the V-I-S (vegetation–impervious surface–soil) model to classify land use in Brisbane, Australia, based on Landsat5 data. The spectrally decomposed coverage images of each class and the V-I-S section results revealed the composition and expansion of Brisbane. Jie

and Wang [5] applied the V-I-S model to extract the impervious surfaces of Hong Kong's Kowloon Peninsula and surrounding urban areas, whereby surface temperature inversion results were used to remove the impervious surface information from the low albedo regions to further improve the classification accuracy (RSME of 10.26%). Yuan [8] extracted information from Landsat imagery of Beijing via the V-H-L-S (vegetation-high albedo-low albedo-soil) model and discussed the ecological impact of imperviousness. Qiu et al. [9] used a B-D-G (brightness–darkness–greenness)-based soil classification model via a simple threshold method on MODIS (moderate-resolution Imaging Spectroradiometer) imagery, with an accuracy of 94%, and kappa index of 0.8789. In conclusion, the majority of related research focus on extracting only impervious surfaces and the subsequent analysis of the urban expansion [10], while studies on the secondary classification of permeable surfaces are lacking.

However, pixels in the middle and coarse spatial resolution image generally combine asphalt, cement pavements, color steel roofs, and other manmade covers, and only one or two IS classes cannot present their physical properties well, especially their heating characteristics. Thus, some scholars have discussed the possibility of subdividing the IS classes. Xiong et al. [11] extracted seven impervious surface classes in Changping, Beijing: cultivated land, woodland, water, bare land, buildings, cement, and asphalt. Zheng et al. [12] took Xian'an, Xian Ning City as the study area to classify land use using buffer analysis, road width, and watershed area with an accuracy of 93.7%. However, they still did not address the mixed pixel problem during the classification.

Some scholars tried to divide the IS into finer classes using hyperspectral images. Ye et al. [13] proposed the extraction of surface material information using hyperspectral images of building materials with varying spectral sensitivities. HRS (High-Resolution Stereoscopic) Hyperion and China's airborne hyperspectral PHI (Push broom Hyperspectral Imager) imagery was used as inputs for small-scale extraction experiments to demonstrate the reliable classification of materials such as color steel tiles, cement concrete, marble, and asphalt. Although Ye was able to classify the urban built-up area in detail, further classification of the city based on the obtained spectral results was not performed. Jilge et al. [14] discussed whether gradient analysis is applicable for mixed decomposition of complex spectra. Similarities of material compositions were analyzed based on 153 systematically distributed samples on a detailed surface material map using detrended correspondence analysis (DCA). The study subdivided IS in Munich, Germany, into 27 subcategories with a simulated Environmental Mapping and Analysis Program (EnMAP) imagery. This method is based on the land cover of German, a developed country, it may not be suitable for the developing country where the building roof types are significantly different from the developed countries.

Although IS was subdivided into finer classes using hyperspectral imageries, the finer IS mapping using multispectral imagery has not been discussed yet. The current class subdivision using multispectral images focused more on the finer vegetation type mapping. Raczko [15] divided the forests of Poland's Szklarska Poręba region into five classes: spruce, larch, shrub, beech, and birch; Ren [16] classified the Baihua forest farm in Gansu into seven classes, with an overall accuracy of 92.28%. Cui and Liu [17] fused spectral information with a random forest method to divide vegetation into five classes: *Artemisia salina*, *Spartina alterniflora*, reed, woodland, and other vegetation types.

The random forest (RF) [18] algorithm is both computationally efficient and accurate in terms of its classification output. It is one of the most commonly used classifiers in recent years, particularly in land classification applications, due to its high accuracy, fast computation speed and excellent stability. For example, Cai [19] employed the random forest method to classify cities based on high score data, with a secondary layer classification accuracy reaching 89%. Due to the high computational efficiency and unlimited number of classifications in spectral mixing analysis, RF outperforms other methods [20].

Therefore, this study aims to explore the feasibility of mapping the finer IS using multispectral image. First, it proposes a classification scheme based on major IS type in



the study area. Second, finer IS classes are mapped in subpixel scale using random forest model with the Landsat 8 imagery in Guangzhou city, China. Finally, root mean square error of each finer IS class is calculated using the estimated sample and according reference values to assess the accuracy of the classification. The results in this study are expected to provide more fundamental information of the inner urban construction which is valuable for the micro-ecology studies and urban management.

## 2. Materials and Methods

### 2.1. Study Area

Guangzhou is located at the estuary of the Pearl River, China. Its latitude and longitude range from 112°57′–114°3′ E, 22°26′–23°56′ N, with a high and low topography in the northeast and northwest, respectively. The north is generally composed of low mountains and hills, while the south is dominated by plains. The city is divided into 11 administrative regions, Baiyun, Conghua, Haizhu, Huadu, Huangpu, Liwan, Nansha, Panyu, Tianhe, Yuexiu and Zengcheng (Figure 1).

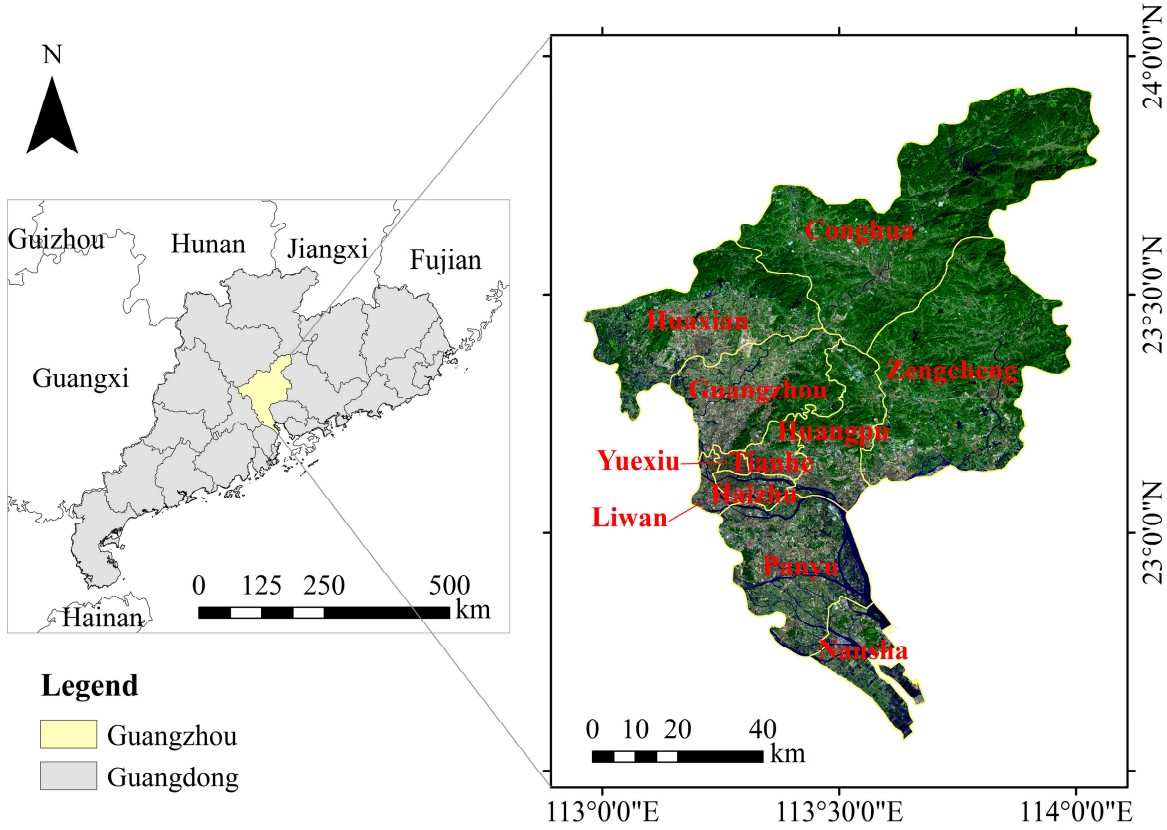

**Figure 1.** Location of the study area, Guangzhou, China.

### 2.2. Datasets and Data Processing

This paper employed Landsat 8 Operational Land Imager (OLI) imagery from geospatial data cloud of Guangzhou city collected on 7 February 2016 (http://www.gscloud.cn/search) at the input for the proposed approach. The image had a spatial resolution of 30 m × 30 m. Pre-processing steps included spectral calibration, atmospheric correction, and the application of vegetation and water masks. The radiometric calibration and atmospheric correction procedures were completed in accordance with the method suggested by the United States Geological Survey (USGS) (http://glovis.usgs.gov/).

The modified normalized difference water index (MNDWI) [21] and normalized difference vegetation index (NDVI) [22] were used to mask the water and vegetation pixels, and are described in Equations (1) and (2), respectively.

$$\text{MNDWI} = \frac{G - MIR1}{G + MIR1},\tag{1}$$

$$\text{NDVI} = \frac{NIR - R}{NIR + R},\tag{2}$$

where $G$, $MIR1$, $NIR$ and $R$ are the green, mid-infrared, near-infrared and red bands, respectively. Figure 2 presents the gray histogram derived via the MNDWI for water and non-water sampling pixels. Based on the histogram, the optimal mask threshold was determined as 0.425. Similarly, the NDVI pixel values were used to mask the vegetation in order to reduce the occurrence of non-vegetation mis-masking [23,24]. The vegetation mask threshold was determined as 0.747, ensuring that non-vegetation pixels were not mis-masked while also reducing the classification error (Figure 3).

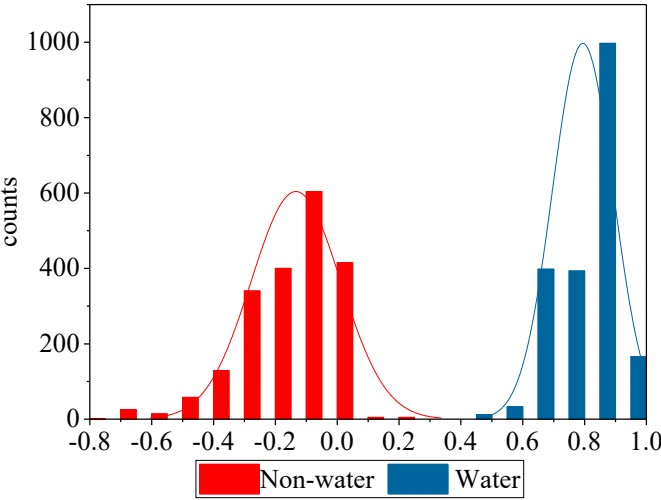

**Figure 2.** Histogram of water and non-water pixels. The x- and y-axis denote the value of the modified normalized difference water index (MNDWI) of each pixel and the number of pixels, respectively.

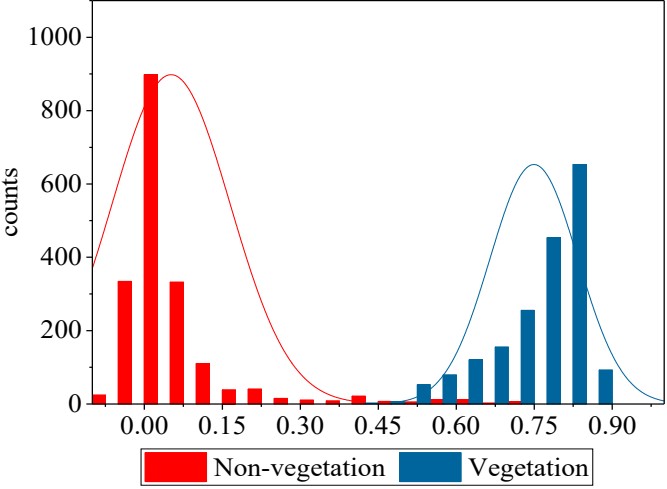

**Figure 3.** Histogram of vegetation and non-vegetation pixels based on normalized difference vegetation index (NDVI) values.

*2.3. Methods*

The classification process is described as follows. First, images with minimal cloud coverage over the study area were selected and underwent preprocessing (Section 2.2). Second, a detailed classification system for a wide range of materials (blue steel, cement, asphalt, other impervious surfaces, other metal, brick, plastic) was determined based on the spectral information of objects with impervious surfaces within the city. This was combined with high-resolution images from Google Earth in order to select samples. These samples were grouped into training (1-pixel 30 m × 30 m) and reference (3-pixel 90 m × 90 m) samples via the established classification system, and the random forest method then performed image mixing decomposition on the sub-pixel scale of the preprocessed images to extract the coverage of the various impervious surface classes in Guangzhou. Stratified random sampling is selected as the sampling method. Firstly, according to the high-resolution historical images provided by Google Earth and the spectral characteristics, the images were classified by visual interpretation, and the impervious surface of Guangzhou was classified in detail. Then we evenly selected each class of samples on the image. Referenced fractions of each class in Google Earth images were extracted through digitizing the corresponding areas within samples. The spatial distribution and area of each class in Guangzhou were determined, and the potential urban planning and urban fine management strategies based on the results are discussed. Figure 4 details the steps of the analysis procedure.

2.3.1. Finer Impervious Surfaces (IS) Classification Scheme

In the initial experiment stage, the researchers reviewed the published papers about land cover in Guangzhou research on its classification [25–28]. Then based on methods for extracting information about the construction materials [13], the detailed classification of impervious surface is confirmed. The effects of artificial land cover on urban climate and ecology vary with type [29]. Composition materials include plastic, metal, rubber, glass, cement, wood, shingles, sand, gravel, brick, stone, etc. The surface material information of ground objects based on the varying spectral sensitivity of different building materials was extracted [13,19]. Data obtained from the field survey of Guangzhou build-up was used to determine the artificial feature classes in each region at a greater spatial resolution. It is determined that the city of Guangzhou is roughly divided into impervious surfaces, bare land, water, woodland, and grassland. Among them, the impervious surfaces are divided into blue steel, cement, asphalt, other impervious surface, other metal, brick, and plastic. A total of 7 classes of ground objects were determined eventually (Figure 5).

2.3.2. Endmember Selection

Ground objects were identified on the Landsat 8 image and combined with historical high-resolution imagery between December 2015 to March 2016 from Google Earth. Each object class with 10 pixels (1-pixel 30 m × 30 m) was used as the training sample. Figure 6 shows the spectrum curve of the selected pure end members. Mixed pixels containing vegetation were still observed in the masked image, thus pure woodland and grassland endmembers were selected in order to improve the classification accuracy.

Following this, reference sample with a dimension of 3 × 3 pixels (90 m × 90 m) were selected. The average value of the pixels in the reference sample window were used to estimate the object coverage in the sample window. The sampling window of the corresponding location and area on the Google Earth high-resolution image were subsequently determined, and the proportion of ground objects in the sampling area was visually estimated. This was used as a reference value for the feature coverage. A total of 103 training and 231 reference sample were determined. Table 1 reports the number of ground object samples.

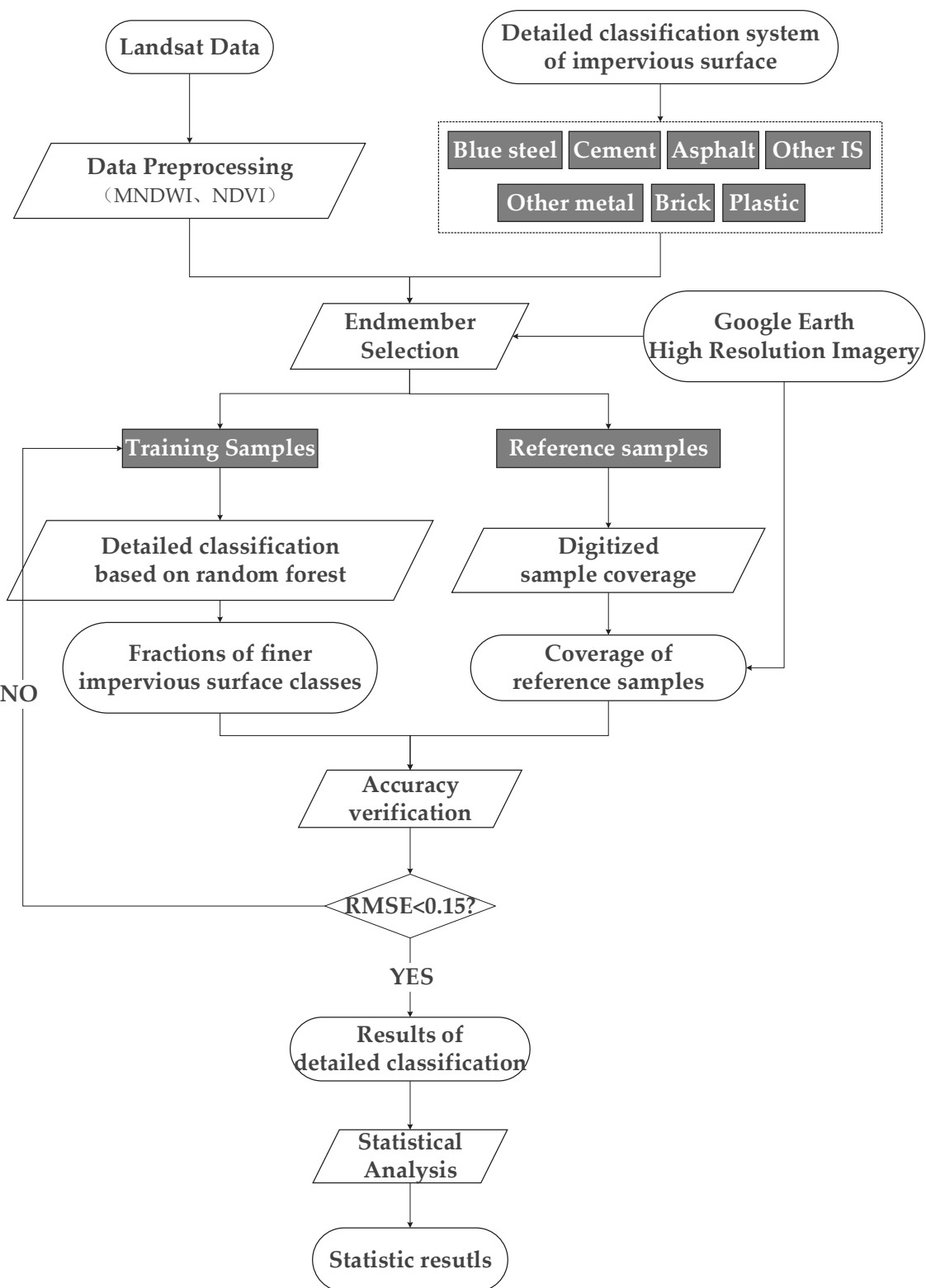

**Figure 4.** Flow chart of classification procedure (MNDWI: modified normalized difference water index; NDVI: normalized difference vegetation index; Other IS: other impervious surface; RMSE: root mean square error).

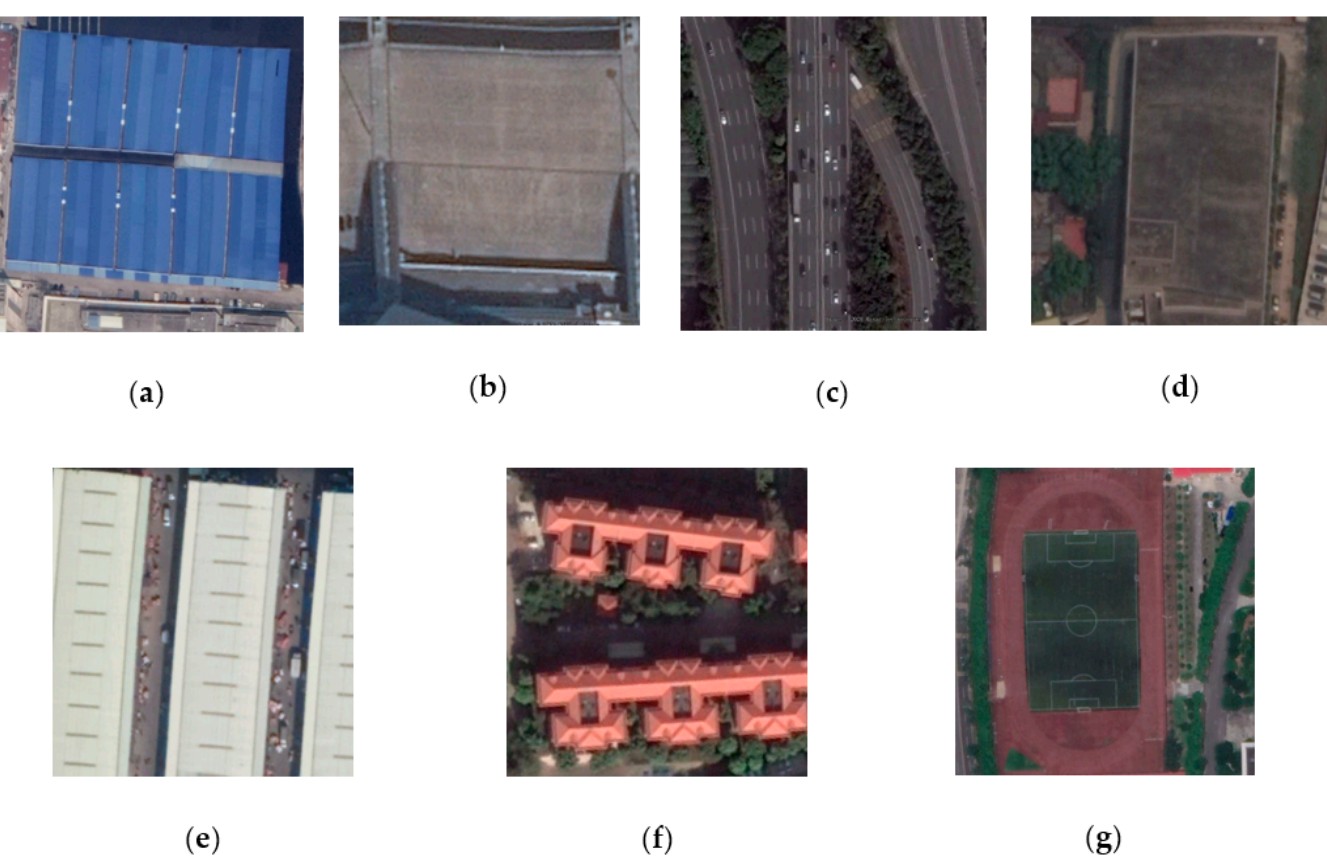

**Figure 5.** Detailed classification of ground objects (**a**) blue steel; (**b**) cement; (**c**) asphalt; (**d**) other impervious surface; (**e**) other metal; (**f**) brick; (**g**) plastic.

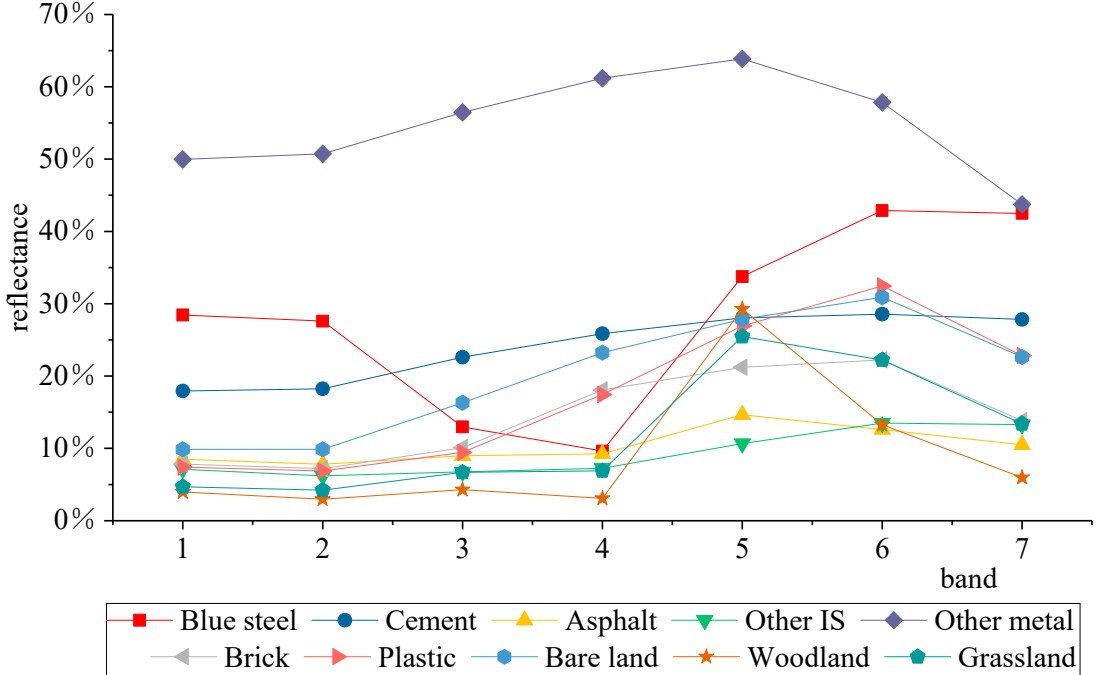

**Figure 6.** Spectral endmembers curves (Other IS: other impervious surface).

**Table 1.** Numbers of samples for each class.

| Classes | Training Samples (1-Pixel, 30 m × 30 m) | Reference Samples (3-Pixel, 90 m × 90 m) |
|---|---|---|
| Blue steel | 23 | 25 |
| Cement | 10 | 35 |
| Asphalt | 10 | 30 |
| Other IS [1] | 5 | 30 |
| Other metal | 8 | 17 |
| Brick | 8 | 30 |
| Plastic | 6 | 30 |
| Other classes [2] | 33 | 34 |
| Total | 103 | 231 |

[1] Other IS: other impervious surface; [2] Other classes: bare land, woodland, and grassland.

### 2.3.3. Random Forest-Based Detailed Classification

The classification approach is based on the random forest method. More specifically, the bootstrap sampling was applied to randomly determine N samples from the original dataset with replacement to form a training set. This process is denoted as bagging. Following this, for each node, m (m ≤ M) features were randomly selected from all M original feature variables and divided into internal nodes. The prediction results of the N decision trees generated by the collection were then determined by voting in order to select the new sample classes [30]. RF has a high prediction accuracy, can handle high-dimensional and multicollinear data, has a high tolerance for outliers and missing values, requires less manual intervention, and is not prone to overfitting problems [31,32]. In this paper, the spectral characteristics obtained from the remote sensing images were used to test the contribution of the feature variables to the classification. This allowed us to select the optimal classification scheme for the detailed urban impervious surface classification. Herein, this study adopted the Random Forest Classifier from the sklearn for classification. The parameters of Random Forest Classifier from scikit-learn are set as: n_estimators = 120, oob_score = True, n_jobs = 2, random_state = 42.

In addition, the support vector machine (SVM) classifier has very effective performance in classification tasks with limited training samples [33]. It was also employed in this study in order to compare the performance of random forest.

### 2.3.4. Accuracy Assessment

The *RMSE* (root mean square error) of the sample reference and estimated values were used as the accuracy verification method for the proposed approach:

$$RMSE = \sqrt{\frac{\sum_{i=1}^{N}(Y_i - X_i)^2}{N}}, \tag{3}$$

where $X_i$ is the estimated value of the coverage of ground objects; $Y_i$ represents the reference value of the ground object coverage; and $N$ is the number of reference sample for each class. The smaller the value of *RMSE*, the better the classification of the model. High-resolution images from Google Earth were employed as the accuracy verification data, and the *RMSE* as the accuracy evaluation index.

## 3. Results

### 3.1. Fractions of Finer Impervious Surface Classes

Since the water and vegetation were masked before the classification, statistical analysis was not performed on the areas with vegetation and water. Figure 7 presents the coverage images of the 7 classes (blue steel, cement, asphalt, other impervious surface, other metal, brick, and plastic). The impervious surfaces are generally located in the west of Guangzhou, particularly in Tianhe and Yuexiu. Among them, the distribution of blue steel

is concentrated principally in the central area of Haizhu and Huadu, and the southwestern area of Baiyun, while cement is mainly concentrated in the southern part of Baiyun. The coverage image of asphalt clearly exhibits the principle traffic arteries of each city circle in Guangzhou. The distribution of the other impervious surfaces is generally concentrated in the four central urban areas of Liwan, Yuexiu, Tianhe, and Haizhu, all of which have higher building densities compared to the other regions. The distribution of the other metal is clustered in industrial areas, such as the southwest of Tianhe, the northwest of Panyu, and the south of Huadu. Brick surfaces are mainly observed in the southwest of Zengcheng, the east of Baiyun, and the southwest of Huangpu. Plastic surfaces are principally concentrated in the universities around Central Lake Park in the northeast of Panyu and schools and large stadiums of Yuexiu, Tianhe, and Haizhu.

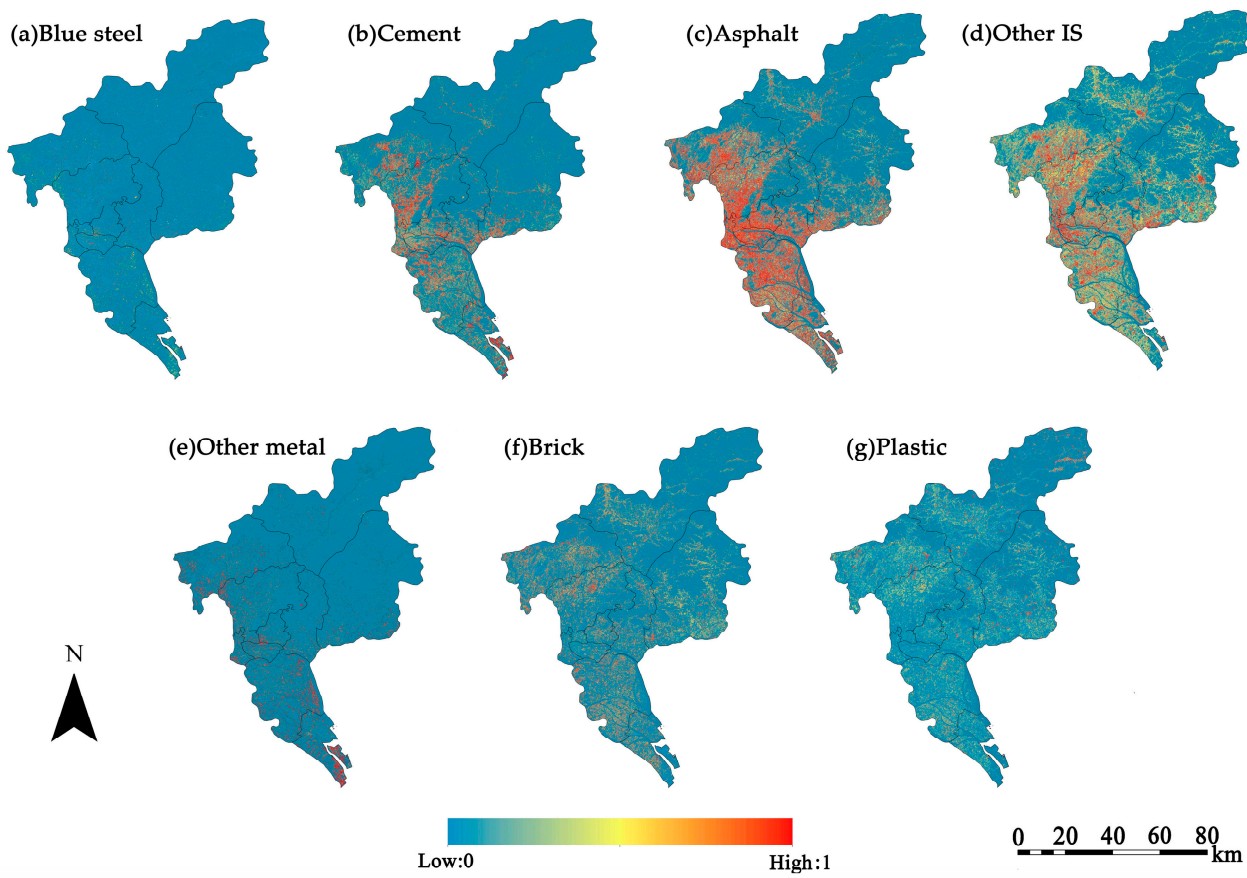

**Figure 7.** Fractions of finer IS classes (**a**) blue steel; (**b**) cement; (**c**) asphalt; (**d**) other impervious surface; (**e**) other metal; (**f**) brick; (**g**) plastic.

The result of impervious surface area (ISA) is shown in Figure 8. Moreover, Figure 8 selects a typical area of impervious surface in Guangzhou, and the reference samples are enlarged to display more details. At the same time, the reference samples of Landsat 8 band4/3/2 were compared with each class of estimated result, and the selection of reference samples of Google earth was showed. It can be seen from the results of (1)–(7) in Figure 9 that the asphalt shows good extraction. Based on visual interpretation and the detailed classification of impervious surface in Guangzhou, the fractional values of reference samples were estimated. Then the referenced fractions of each class in Google Earth images were extracted through digitizing the corresponding areas within samples (See subfigures C in Figure 9).

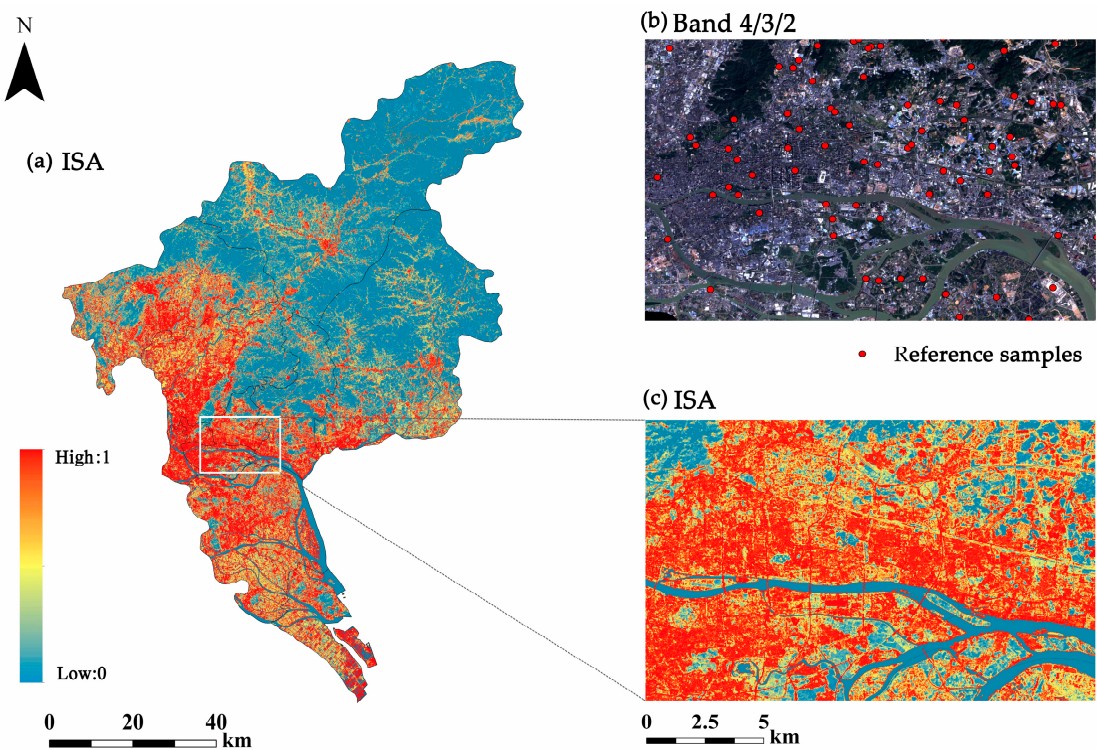

**Figure 8.** Estimated results of ISA. (ISA: impervious surface area; (**a**) ISA of Guangzhou; (**b**) Reference samples on Landsat 8 (Band4/3/2) in representative urban areas of IS; (**c**) ISA of representative urban areas).

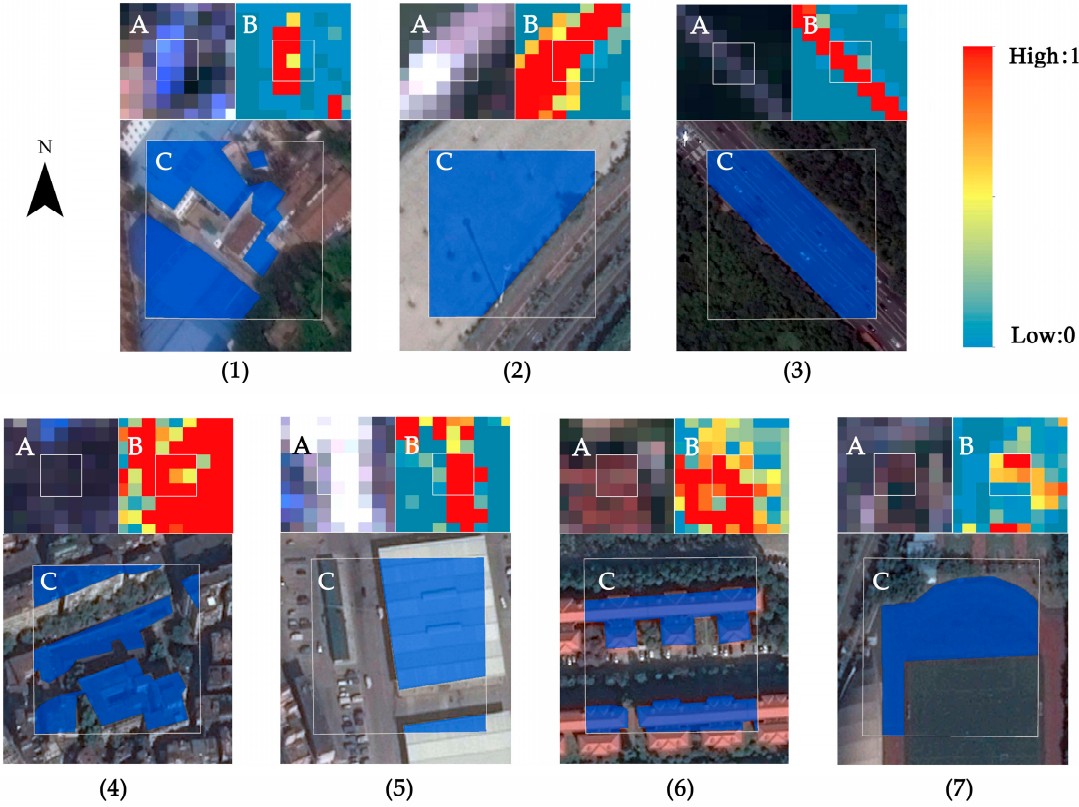

**Figure 9.** Selected reference sample of each IS type. (**1**) blue steel; (**2**) cement; (**3**) asphalt; (**4**) other impervious surface; (**5**) other metal; (**6**) brick; (**7**) plastic; (**A**) Reference samples on Landsat 8 (Band4/3/2); (**B**) Estimated detailed classification results of ISA; (**C**) Reference samples on Google earth, each IS area was highlighted in blue.

*3.2. Accuracy*

The RMSE of each random forest classification class was determined as follows: blue steel 11.75%, cement 9.92%, asphalt 7.99%, other impervious surfaces 10.04%, other metals 12.95%, brick 11.16%, and plastic 8.48% (Figure 10). Thus, asphalt and other metals were observed to have the highest and lowest classification accuracies, respectively. Otherwise, the classification accuracy RMSE of the results obtained by the support vector machine method is: blue steel 12.94%, cement 22.06%, asphalt 11.39%, other impervious surfaces 19.87%, other metals 24.63%, brick 11.63%, and plastic 9.55%. The class with the lowest classification accuracy of the two methods is other metals. It is obvious that the RMSE of all classification class in the RF is lower than that of the support vector machine, especially for other metals, cement and other impervious surfaces. In general, of the two methods, RF has better accuracy for fine classification of cities.

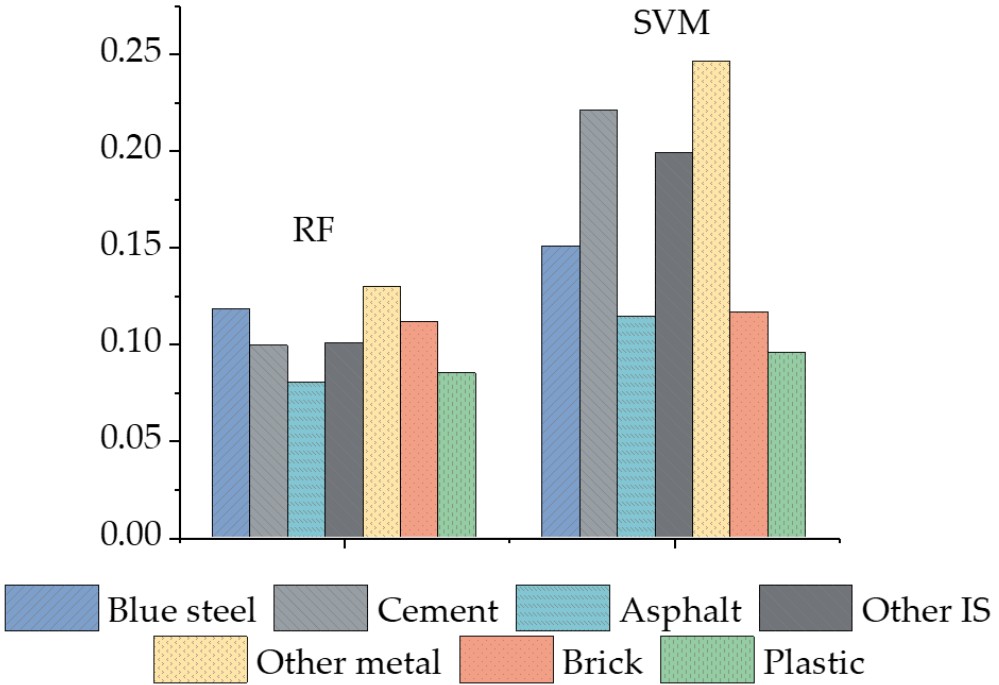

**Figure 10.** RMSE results of each class compared with random forest (RF) and support vector machine (SVM).

The linear relationship between the test values and the estimated values compared with RF and SVM is displayed in Figure 11. Among the selected reference samples, linear regression was used to generate coefficient of determination values. The determination values closer to 1 indicate a better simulation effect of the model. It can be seen from Figure 11 that the RF outperforms the SVM on all seven classifications in terms of accuracy. However, in blue steel and other metal, there was a small difference between the two methods on average. Since the performance of SVM is far inferior to RF, subsequent research mainly uses the result data obtained by RF.

*3.3. Statistic Results*

Table 2 reports the area statistics of each class. The total area of impervious surfaces in Guangzhou is determined as 2258.5 km$^2$, accounting for 36.33% of the total. Figure 12 presents the proportion of each class. Among them, asphalt occupies the largest area in Guangzhou with 691.71 km$^2$ (9.68%), followed by other impervious surfaces, with an area of 447.84 km$^2$ (6.27%). Blue steel occupies the least area in Guangzhou, with a total area of 78.79 km$^2$, accounting for 1.1%.

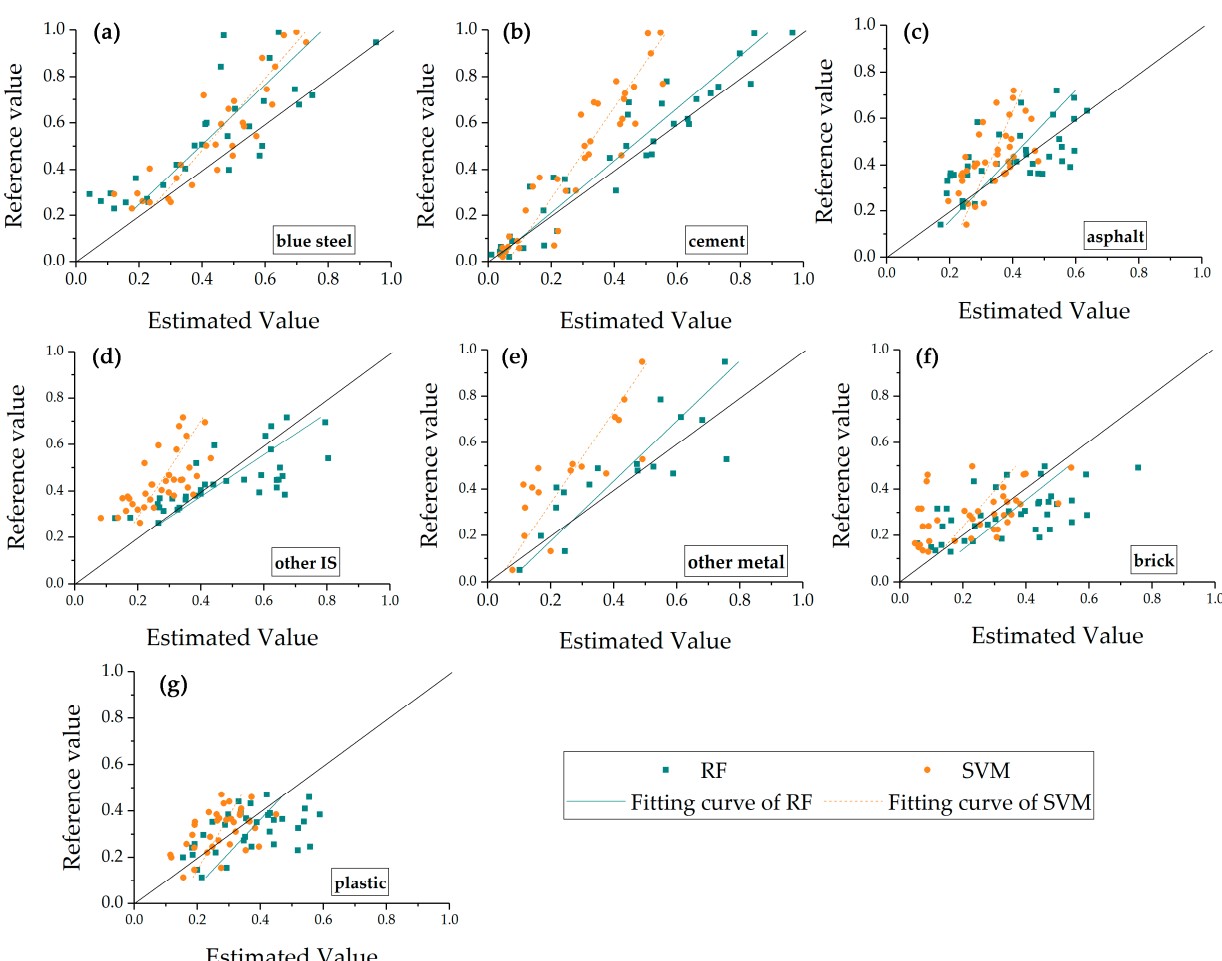

**Figure 11.** Linear relationship between the test values and the estimated values compared with RF and SVM (**a**) blue steel; (**b**) cement; (**c**) asphalt; (**d**) other impervious surface; (**e**) other metal; (**f**) brick; (**g**) plastic.

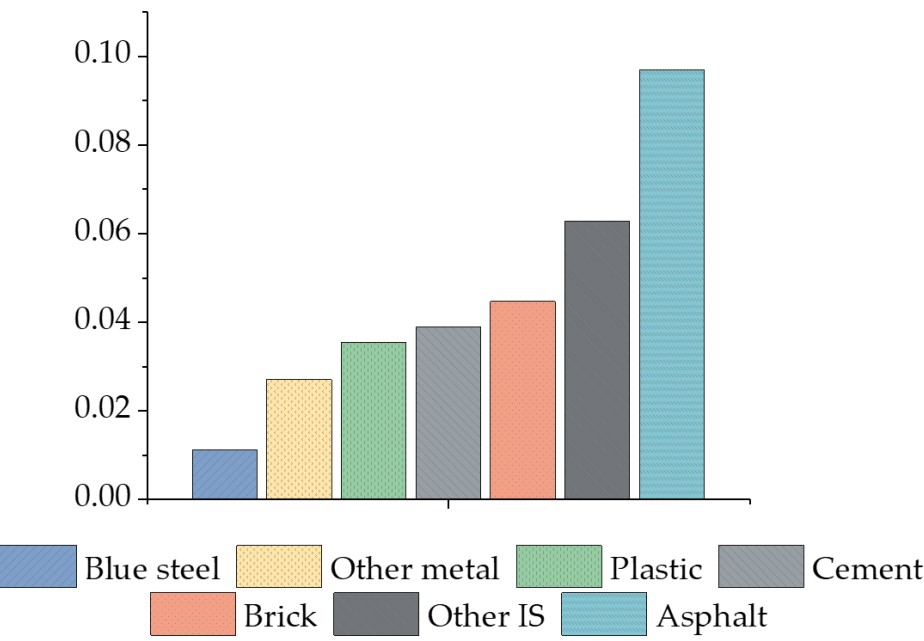

**Figure 12.** Proportion of ground objects in Guangzhou.

**Table 2.** Area of each class in Guangzhou.

| Class | Area (km$^2$) |
|---|---|
| Blue steel | 78.79 |
| Cement | 276.72 |
| Asphalt | 691.71 |
| Other IS [1] | 447.84 |
| Other metal | 192.31 |
| Brick | 318.43 |
| Plastic | 252.72 |
| Guangzhou | 7147.81 |

[1] Other IS: other impervious surface.

Table 3 reports the statistics of the object areas in the 11 administrative districts of Guangzhou. The area distribution of blue steel is concentrated in Panyu, Huadu, Zengcheng, Baiyun, and Conghua, with the largest area in Panyu (14.72 km$^2$) and the lowest in Yuexiu (1.03 km$^2$). Figure 13 identifies Haizhu, Liwan, and Yuexiu to have the highest proportion of blue steel (between 2% and 3%). The distribution of cement is concentrated in Panyu, Huadu, Zengcheng, Baiyun, and Conghua, with a peak in Panyu (58.27 km$^2$) and minimum in Yuexiu (3.07 km$^2$). The proportion of cement is the highest among the three districts of Haizhu, Liwan, and Yuexiu, ranging from 9% to 12%. Asphalt is concentrated in the four districts of Panyu, Huadu, Zengcheng, and Baiyun, with Panyu holding the largest area (163.4 km$^2$) and Yuexiu the lowest (10.01 km$^2$). Furthermore, asphalt accounts for the highest proportion in Haizhu, Liwan, Tianhe, and Yuexiu (20–30%). The other impervious surfaces are principally located in the four districts of Panyu, Huadu, Zengcheng, and Baiyun, with Panyu and Yuexiu exhibiting the largest and lowest distributions of 79.04 km$^2$ and 5.5 km$^2$, respectively. Liwan and Yuexiu exhibit the highest proportion of other impervious surfaces, with values ranging from 14% to 16%. Other metals are distributed in Panyu, Huadu, Zengcheng, and Conghua, with the largest distribution in Panyu (47.65 km$^2$) and lowest in Yuexiu (2.01 km$^2$). The other metals account for the highest proportion in Haizhu, Liwan, Nansha, Panyu, and Yuexiu, with values between 6–9%. Brick surfaces are generally distributed in Panyu, Huadu, Zengcheng, Baiyun, and Conghua, with Zengcheng and Yuexiu exhibiting the largest and lowest distributions of 61.32 km$^2$ and 1.69 km$^2$, respectively. The proportion of bricks is relatively evenly spread across the 11 districts, with values between 2% and 7%. The distribution of plastics is concentrated in the four districts of Panyu, Huadu, Zengcheng, and Conghua, with the greatest area in Conghua (58.3 km$^2$) and the lowest in Yuexiu (0.87 km$^2$). The proportion of plastics is also evenly distributed within the 11 districts, ranging from 2% to 5%.

**Table 3.** Distribution and area of each class in the districts of Guangzhou.

| District Name | District Area | Blue Steel | Cement | Asphalt | Other IS | Other Metal | Brick | Plastic |
|---|---|---|---|---|---|---|---|---|
| Baiyun | 671.24 | **11.12** | **50.61** | **106.52** | **73.46** | 22.84 | 41.25 | 27.59 |
| Conghua | 1981.24 | 9.58 | 13.72 | 44.19 | 46.08 | 8.99 | 47.73 | **58.30** |
| Haizhu | 100.82 | 3.18 | 9.60 | 26.52 | 15.11 | 6.77 | 4.98 | 2.53 |
| Huadu | 955.20 | **12.63** | **42.03** | **108.79** | 73.46 | **31.69** | **57.06** | **41.45** |
| Huangpu | 457.11 | 5.17 | 22.71 | 48.57 | 33.35 | 9.92 | 21.58 | 15.46 |
| Liwan | 70.88 | 2.01 | 9.06 | 20.47 | 11.15 | 5.28 | 3.42 | 1.83 |
| Nansha | 336.22 | 5.83 | 27.89 | 53.50 | 21.55 | **31.93** | 18.00 | 10.92 |
| Panyu | 798.75 | **14.72** | **58.27** | **163.40** | **79.04** | **47.65** | **53.63** | 33.22 |
| Tianhe | 127.64 | 2.62 | 10.47 | 26.78 | 15.04 | 4.72 | 7.78 | 4.66 |
| Yuexiu | 32.89 | 1.03 | 3.07 | 10.01 | 5.50 | 2.01 | 1.69 | 0.87 |
| Zengcheng | 1615.81 | 10.90 | 29.29 | 82.96 | **74.11** | 20.51 | **61.32** | **55.88** |

Bold values are the top three regions that occupy the most area in each class.

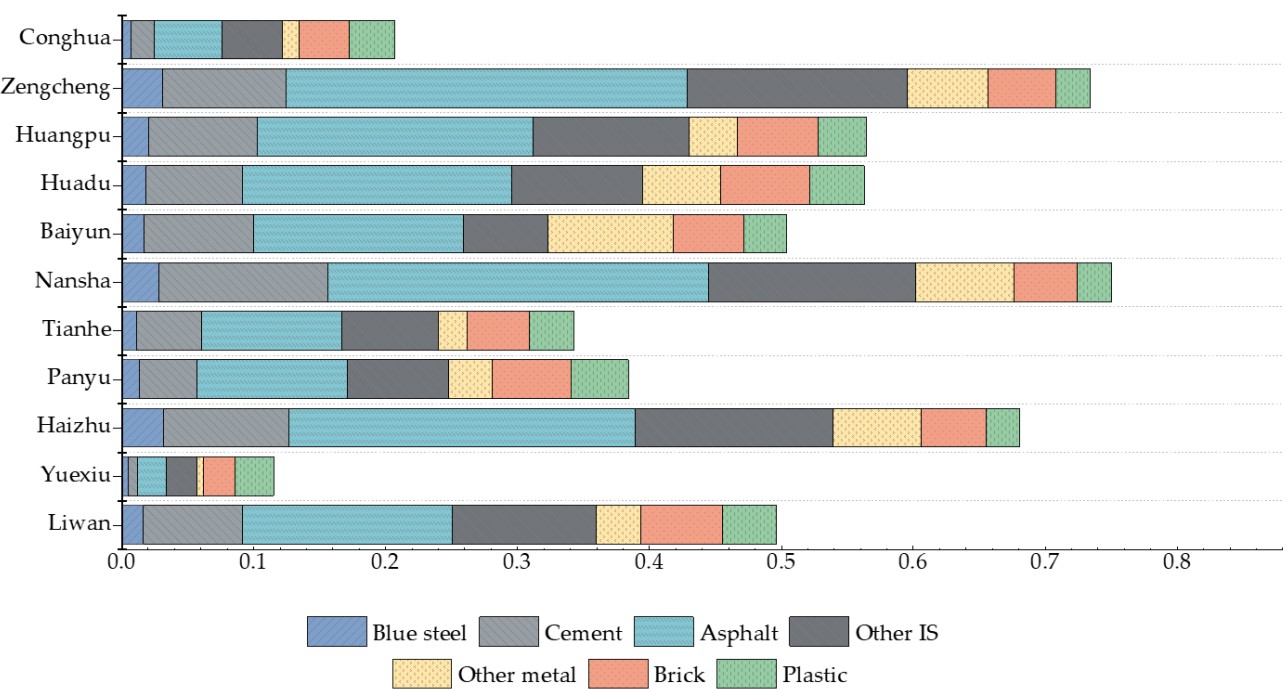

**Figure 13.** Proportion of each class in each district of Guangzhou.

## 4. Discussion

The majority of current related research provides a general classification of all ground objects in urban built-up areas as impervious surfaces. However, this paper presents an approach that classifies impervious surfaces in detail via the application of non-linear spectral hybrid analysis to subdivide impervious surfaces. This makes it possible to determine quantitative information on detailed objects in built-up urban areas. Results provide data for the improvement of urban ecological spaces and confirm that the detailed classification of urban built-up areas can aid in the development of urban planning and policy formulation, thus contributing to future urban development [19].

Previously, some scholars have used random forest and spectral mixture analysis (SMA) to extract the impervious surface, but they have not conducted in-depth analysis of finer impervious surface classes [3,34,35]. They simply divide the impervious surface into one or two classes (high albedo-low albedo) on the basis of different spectral reflectance. This paper used the same methods and data sources as the above research, but the difference is that the detailed classes in the impervious surface was subdivided, and the results achieved reliable accuracy. In addition, some scholars have made a fine classification of cities, but the data sources were different [13,14,19]. Due to the abundant hyperspectral bands, hyperspectral data have obvious advantages in subdivision of ground objects. However, hyperspectral data is computation-intensive and time-consuming, and the research based on it cannot cover a large area. For example, Zhong et al. [36] used unmanned aerial vehicle (UAV)-borne hyperspectral systems to acquire hyperspectral imagery for precise crop classification. This research was focusing on agricultural instead of urban structures.

The detailed classification of impervious surfaces not only improves the extraction accuracy of impervious surfaces in low and medium resolution data, but also provides quantitative statistics on the internal structure of urban built-up areas and a wide range of objects in built-up areas. The classification of blue steel and other metals is useful for research on the distribution of industrial areas in Guangzhou and variations in the urban temperature [37]. The extraction of cement presented here is of positive significance to research on urban underlying surfaces [38]. The classification results of asphalt are also important for the planning of urban road and maintains the urban environment [39], for example, common road and roofing asphalts produced complex mixtures of organic com-

pounds, including hazardous pollutants [40]. The classification results of other impervious surfaces and bricks can provide reference data for studies on the spatial distribution of residential areas in Guangzhou [41]. Changes in plastic and its spatial distribution can also contribute to the division of urban functional areas such as schools and stadiums. Recently, Elhacham et al. [42] have quantified the human-made mass, referred to as 'anthropogenic mass', which include plastic, metal, glass, cement, brick, etc., and compare it to the overall living biomass on Earth. The result showed that for each person on the globe, anthropogenic mass equal to more than his or her bodyweight is produced every week. Similar to their purpose, quantifying the artificial ground objects has become an important step in investigating the impact of human development on nature.

The detailed classification of impervious surfaces also has an important impact on the internal temperature, ecology and urban planning of cities [43]. Detailed classifications are of great research significance for the effective mitigation of the heat island effect, as well as quantifying the specific classes of surface coverage in urban areas, including the detailed composition of materials (asphalt roads, non-ferrous metals, bright and dark roofs, etc.) [29,44]. A wide range of ground objects with impervious surfaces have altered the urban ecological spatial structure. A detailed classification can provide effective data for urban ecological carrying capacity and urban greening planning strategies [45,46]. Moreover, ground objects with impervious surfaces also have different service lives and aging rates. Research on their distribution has a positive significance on population production, city appearance maintenance, disaster emergency management, and urban village reconstruction.

In addition, the image was collected during winter in the northern hemisphere, and part of the arable land exhibits the spectral characteristics of bare land due to autumn harvest and the newly expanded building space. Consequently, the classification results demonstrate the presence of bare land near Nansha in the southwest of the study area. Although the image was subject to water and vegetation masks prior to the classification, some water and vegetation pixels were still present. In particular, for the vegetation mask, the model easily confuses dark blue steel with vegetation, resulting in incorrect masking results.

The RMSE of blue steel, other impervious surfaces, other metals, and brick are all greater than 10%. The classification result of the above classes is not precise enough. Similar to this study findings, some scholars have proposed that due to the similar spectral characteristics, blue steel is easy to confuse with shadows, low reflection buildings and vegetation [47,48]. Meanwhile, Li, et al. [49] have demonstrated that buildings are commonly found with complex spectral and spatial characteristics in densely populated urban areas. Shadow and other impervious surfaces may have similar spectral, textual, and geometrical characteristics, resulting in a high mis-classification rate between them. Moreover, Iftene, et al. [50] proved that other metal and light roof buildings cannot be distinguished well in extraction. However, he found that the height characteristics can be used to distinguish between the two ground objects. At the same time, he proposed that with the aid of height information or data such as shadows and light detection and ranging (LiDAR), the metal extraction results will be better. Furthermore, Yan, et al. [51] proposed that the brick is similar to that of bare land in the visible spectrum, but in the mid-infrared band, the brightness of the sun-facing side of the brick is significantly higher than that of the bare land and the shaded brick. From their research results, it can be found that mid-infrared band is very effective for the extraction of bricks.

This study focuses on distinguishing different finer impervious surface classes in urban area with the random forest model. The random forest model is flexible for different data acquired from various remote-sensing platforms. It can be applied to different remotely sensed data with their according samples. The major challenge of this study is selecting the samples of each finer impervious surface class. Coarse spatial resolution imagery's pixels cover a large ground area, and it is hard to select pure finer impervious surface class samples (a pixel only contains one finer impervious surface class). For example,

the image of moderate-resolution imaging spectroradiometer (MODIS), its best spatial resolution is 250 m, which it covers $250 \times 250 = 62,500$ square meters in a pixel. Many finer impervious surface classes in urban areas have areas of far less than 62,500 squares. Thus, it is hard to collect these samples, preventing the application of finer impervious surface classes identification. The samples recorded from the field investigation and lab experiment, to some degree, can address the limitation of sample selection. However, high-quality atmospheric correction should be applied to the imagery before classification to avoid the mismatch between the spectra recorded from the remotely sensors and the samples recorded from ground experiments. For the imageries which have finer spatial resolution than Landsat data, the method in this study can be perfectly applied since these types of imageries can present more pure finer impervious surface classes in a pixel [52]. The limitation of selecting pure samples can be easily addressed in higher spatial resolution imageries. Therefore, this method can be applied to higher spatial resolution (spatial resolution is higher than 30 m) imagery.

## 5. Conclusions

This study extracted the finer IS classes using Landsat imagery with a random forest method. The IS was divided into seven finer impervious surface classes (blue steel, cement, asphalt, other impervious surfaces, other metal, brick, and plastic). Several conclusions can be drawn as follows:

(1) Finer impervious surface classes can be divided using the random forest classification method within Landsat data. RMSE values of all impervious surface classes are below 15%, with asphalt demonstrating the highest classification accuracy.

(2) The total area of impervious surfaces in the study area is 2258.5 km$^2$, accounting for 36.33% of the entire Guangzhou. Asphalt, other impervious surface, and brick are the dominant impervious surface area types with the percentages of 9.68%, 6.27%, and 4.45%, respectively. They are mainly located in Yuexiu, Liwan, Haizhu, and Panyu districts.

This study is a trial to distinguish the finer impervious surface classes using median spatial resolution images. Its major contribution is providing a more detailed structure information about the urban areas which could be used for urban toughness analysis [53–55], urban micro ecology [56] and urban planning [57].

**Author Contributions:** Conceptualization, W.L. and Y.D.; methodology, W.L. and Y.D.; software, Y.D.; validation, W.L.; formal analysis, W.L.; investigation, W.L. and M.S.; writing—original draft preparation, W.L.; writing—review and editing, W.L., Y.D. and M.L.; data curation, J.Y. and J.X.; All authors have read and agreed to the published version of the manuscript.

**Funding:** This research was funded by National Natural Science Foundation of China, grant number 41901372 and 41901072, Key Special Project for Introduced Talents Team of Southern Marine Science and Engineering Guangdong Laboratory (Guangzhou) [grant number GML2019ZD0301], Science and Technology Program of Guangzhou [grant number 202002030247], GDAS Project of Science and Technology Development [2019GDASYL-0103004].

**Institutional Review Board Statement:** Not applicable.

**Informed Consent Statement:** Not applicable.

**Data Availability Statement:** The data presented in this study are available on request from the corresponding website.

**Acknowledgments:** We would like to thank the anonymous reviewers for their constructive comments.

**Conflicts of Interest:** The authors declare no conflict of interest.

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
