# Peer review of "Extraction and Analysis of Finer Impervious Surface Classes in Urban Area"

_remotesensing, doi:10.3390/rs13030459_

Round 1

Reviewer 1 Report

Analysing this proposed article, I have the following suggestions and comments:

The title of this paper is well correlated with its contents. However, I suggest to replace „urban environment” with „urban area”, which seems to be more appropriate. The content are more elements regarding the urban structure and less regarding the individual influence of each analysed class, and the synergism of all classes on the quality of urban environment.

The Abstract reflects the main findings of the reesearch, and offers relevant elements to arouse the interest of the reader. Please, clarify why in the Abstract there are mentioned seven IS classes, and inside of the content eight classes!

The Introduction comprises relevant presentation of the literature in the field, with the most interesting findings and methdos used by different reserchers, in the last years. I believe that this section is significant, demonstrating the importance of debates and research on the finer impervious surfaces in urban area.

I highly appreciate the second section of the paper, Materials and Methods, which is very well structured and very well described to be understood and applied by other scholars in their studies. It is a real pattern for similar analysis. It is enough to follow the steps presented in the Fig. 4, to appreciate the logical approach of the research. For each step, the authors not exaggerate with details, specifying sources of images, the pre-processing steps, used indexes, ry, the  importance of correlation between Landsat Data and Google Earth Imagery... until the validation of the finer impervious surface classes. The statistical approach is correct and convincing.

The Results and Discussion sections are relevant and validate the used method to extract the eight finer imprevious surface, demonstrating the advantages of this methodological way to depict the territorial distribution of these fractions in Guangzhou City. No comments regarding both excellent contents!

The Conclusions section seems to be less developed! There is a discrepancy between the excellent analysis and this very short conclusions. Surely the authors are able to add more relevant methodological issues, which could be developed in further reesearch. Using the obtained results, the authors can comment the potential importance of the finer impervious surface distribution for applied research, oppening specific debates about how these results could be used for urban resilience increase (putting the accent on the urban micro-ecological structures), for example.  

I recommend the acceptance for publication of this paper, with minor changes.

Author Response

Many thanks for the constructive comments on this manuscript. We have addressed all the suggested comments one by one, and the details are listed as follows.

Point 1: It varies by scientists and journals but the use of personal pronouns (we, our) is not typical in scientific text. The title of this paper is well correlated with its contents. However, I suggest to replace „urban environment” with „urban area”, which seems to be more appropriate. The content are more elements regarding the urban structure and less regarding the individual influence of each analysed class, and the synergism of all classes on the quality of urban environment.                 

Response 1: Many thanks for your suggestions. 1) We have corrected all the personal pronouns in this manuscript. 2) the term area is better to capture the main point of our study. We have corrected the title to “Extraction and Analysis of Finer Impervious Surface Classes in Urban Area”.

Point 2: The Abstract reflects the main findings of the research, and offers relevant elements to arouse the interest of the reader. Please, clarify why in the Abstract there are mentioned seven IS classes, and inside of the content eight classes!

Response 2: We are sorry for causing this confusion. At first, there are 7 impervious surface classes and one natural land cover type (soil) were applied to the finer classification. We did not present well in the abstract. Now we have deleted the soil class and modified the corresponding pictures (Figure 5, Figure 7, Figure 10, Figure 12, and Figure 13) to make it clearer. Meanwhile, we have corrected these problems in Lines 229-232.

Point 3: The Conclusions section seems to be less developed! There is a discrepancy between the excellent analysis and this very short conclusions. Surely the authors are able to add more relevant methodological issues, which could be developed in further research. Using the obtained results, the authors can comment the potential importance of the finer impervious surface distribution for applied research, opening specific debates about how these results could be used for urban resilience increase (putting the accent on the urban micro-ecological structures), for example.

Response 3: Thanks for these constructive feedbacks. This study is a trial to distinguish the finer impervious surface classes using median spatial resolution images. It is the previous study of analysing the urban micro-ecological which will use these finer impervious surface classes. Thus, we extend the direction of our study by adding some potential applications of using finer impervious surface. In the last paragraph of the conclusion section, we added: “This study is a trial to distinguish the finer impervious surface classes using median spatial resolution images. Its major contribution is providing a more detailed structure information about the urban areas which could be used for urban toughness analysis [53-55], urban micro ecology[56] and urban planning[57]. (Please check Lines 442-445)

Reviewer 2 Report

It is appreciated that the manuscript is easy to follow and not too long. With interest I read the manuscript, as the message is clear and of interest to the community. The authors tried to shed some light on an important issue pertaining to Impervious surfaces (IS) by proposing a finer IS classification scheme and mapping the detailed IS fraction in Guangzhou City.

Proposed method seemed to be promising in terms of computational simplicity and classification accuracy. However, I would suggest major revision before the manuscript could be accepted. Please allow me to clarify.

Line 117: Was there any specific reason why a low spatial resolution satellite imagery was considered, provided there are other high resolution satellite images such as Sentinel-2 are freely available?

Line 199: There is plenty of literature suggesting the efficacy of random forest based classification provided there are enough training samples available. However, support vector machine based classification is found to be more efficient when number of training samples are less. Having said that, it would be interesting to witness in the revised version how random forest results are comparable to support vector machine and other state of the art classification methods.

Line 204: What parameter values of the random forest classification method was used and how they were determined?

Could you also include in your discussion section about how this methodology could be extended for other available satellite based multispectral sensors? And, what are the expected challenges and advantages do you expect exploiting them?

Author Response

Many thanks for the constructive comments on this manuscript. We have addressed all the suggested comments, and the details are listed as follows.

Point 1: Line 117: Was there any specific reason why a low spatial resolution satellite imagery was considered, provided there are other high resolution satellite images such as Sentinel-2 are freely available?

Response 1: Yes, we have reasons for why we chose the Landsat instead of the Sentinel-2. This study is a trial to distinguish the finer impervious surface classes using median spatial resolution images. Our future objective is to extract the finer impervious surface classes in large scale study area (e.g., Guangdong-Hong Kong-Macao Greater Bay Area) and lone time series spatial-temporal change analysis.  Landsat data are available from 1970s to present, which an ideal dataset to review the detailed change of the larger scale from a lone time. This is why we select the Landsat data as the data source instead of the Sentinel-2 which is only available from 2015.

Point 2: Line 199: There is plenty of literature suggesting the efficacy of random forest based classification provided there are enough training samples available. However, support vector machine based classification is found to be more efficient when number of training samples are less. Having said that, it would be interesting to witness in the revised version how random forest results are comparable to support vector machine and other state of the art classification methods.

Response 2: Many thanks for this suggestion. 1)We added the experiment of SVM for comparison in this study. 1) Descriptions of SVM was added in the method section 2.3.3 in Lines 216-218. 2)We compared their accuracies in section 3.2 using root mean square error (RMSE)(Lines 268-277). 3) Results showed that RF performed better than SVM in extracting finer impervious surface classes in urban as the RMSEs of RF are smaller than the RMSEs of SVM (Lines 268-288).

Point 3: Line 204: What parameter values of the random forest classification method was used and how they were determined?

Response 3: We are sorry for missing these paremeters in our model. The number of estimators in random forest is 120, the oob_score is set as true, the number of jobs is two which matchs the CPU of our computer, and the other parameters are set as default. We have added the parameter values of the random forest classification method in Lines 212-215 as “Herein, this study adopted the Random Forest Classifier from the sklearn. The parameters of Random Forest Classifier from scikit-learn are set as : n_estimators=120, oob_score = True, n_jobs=2,random_state=42.

Point 4: Could you also include in your discussion section about how this methodology could be extended for other available satellite based multispectral sensors? And, what are the expected challenges and advantages do you expect exploiting them?

Response 4: We really appreciate this comment. We added one more paragraph in the section of discussion to discuss the possibility of applying this method to different remotely sensed data. Detail can be seen from Line 408-428:

This study focuses on the distinguishing different finer impervious surface classes in urban area with the random forest model. Random forest model is flexible for different data acquired from various remote sensing platforms. It can be applied to different remotely sensed data with their according samples. The major challenge of this study is selecting the samples of each finer impervious surface class. Coarse spatial resolution imagery’s pixel covers a large ground area, and it is hard to select pure finer impervious surface class samples (a pixel only contains one finer impervious surface class). For example, the image of moderate-resolution imaging spectroradiometer (MODIS), its best spatial resolution is 250m, which it covers 250*250=62500 square meters in a pixel. Many finer impervious surface classes in urban areas its area is far less than 62500 squares. Thus, it is hard to collect these samples, preventing the application of finer impervious surface classes identification. The samples recorded from the field investigation and lab experiment, to some degree, can address the limitation of sample selection. But high-quality atmospheric correction should be applied to the imagery before classification to avoid the mismatch between the spectra recorded from the re-motely sensors and the samples recorded from ground experiments. For the imageries which have finer spatial resolution than Landsat data, the method in this study can be perfectly applied since these types of imageries can present more pure finer impervious surface classes in a pixel [52]. The limitation of selecting pure samples can be easily addressed in higher spatial resolution imageries. Therefore, this method can be applied to higher spatial resolution (spatial resolution is higher than 30 meter) imagery.”  

Round 2

Reviewer 2 Report

Nice work on addressing my concerns and comments. I appreciate the effort and recommend for publication.